# PolyNet: Learning Diverse Solution Strategies for Neural Combinatorial Optimization

**André Hottung**
Bielefeld University, Germany
`andre.hottung@uni-bielefeld.de`

**Mridul Mahajan**
Boston University, USA
`mridulm@bu.edu`

**Kevin Tierney**
Bielefeld University, Germany
`kevin.tierney@uni-bielefeld.de`

## Abstract

Reinforcement learning-based methods for constructing solutions to combinatorial optimization problems are rapidly approaching the performance of human-designed algorithms. To further narrow the gap, learning-based approaches must efficiently explore the solution space during the search process. Recent approaches artificially increase exploration by enforcing diverse solution generation through handcrafted rules, however, these rules can impair solution quality and are difficult to design for more complex problems. In this paper, we introduce PolyNet, an approach for improving exploration of the solution space by learning complementary solution strategies. In contrast to other works, PolyNet uses only a single-decoder and a training schema that does not enforce diverse solution generation through hand-crafted rules. We evaluate PolyNet on four combinatorial optimization problems and observe that the implicit diversity mechanism allows PolyNet to find better solutions than approaches that explicitly enforce diverse solution generation.

## 1 Introduction

There have been remarkable advancements in recent years in the field of learning-based approaches for solving combinatorial optimization (CO) problems (Bello et al., 2016; Kool et al., 2019; Kwon et al., 2020). Notably, reinforcement learning (RL) methods have emerged that build a solution to a problem step-by-step in a sequential decision making process. Initially, these construction techniques struggled to produce high-quality solutions. However, recent methods have surpassed even established operations research heuristics, such as LKH3, for simpler, smaller-scale routing problems. Learning-based approaches thus now have the potential to become versatile tools, capable of learning specialized heuristics tailored to unique business-specific problems. Moreover, with access to sufficiently large training datasets, they may consistently outperform off-the-shelf solvers in numerous scenarios. This work aims to tackle some of the remaining challenges that currently impede the widespread adoption of learning-based heuristic methods in practical applications.

A key limitation of learning-based approaches is that they struggle to sufficiently explore the solution space. The metaheuristics literature identifies exploration as a key component of heuristic optimization procedures (Blum & Roli, 2003), but examining diverse solutions alone is not enough to find high-quality solutions. Mahfoud (1995) discuss the need for *useful diversity*, i.e., diversity that "helps in achieving some purpose or goal." To encourage search diversity, many recent neural construction methods (e.g., Li et al. (2023a); Choo et al. (2022)) follow the POMO approach (Kwon et al., 2020) and improve exploration by forcing diverse first actions during solution construction. While this creates significant diversity through exploiting symmetries in the problem representation (every solution has a different first action), no effort is made to encourage diversity subsequent to the first action. Furthermore, in more complex optimization problems the initial action can significantly influence the solution quality, rendering these methods less effective at generating solutions.

Acknowledging that useful diversity is an essential component of search techniques, neural CO approaches have begun attempting to encourage more exploration during search. Xin et al. (2021)

propose a transformer model with multiple decoders that encourage each decoder to learn a distinct solution strategy during training by maximizing the Kullback-Leibler (KL) divergence between decoder output probabilities. However, to manage computational costs, diversity is only promoted in the initial construction step. In contrast, Grinsztajn et al. (2023) introduce Poppy, a training procedure for multi-decoder models that increases diversity without relying on KL divergence. Poppy generates a population (i.e., a set) of decoders during the learning phase by training only the best-performing decoder for each problem instance. While effective, Poppy is computationally intensive, requiring a separate decoder for each policy, thus limiting the number of learnable policies per problem.

In this paper, we introduce PolyNet to address the previously discussed limitations:

1. PolyNet learns a diverse and complementary set of solution strategies for improved exploration using a single decoder.
2. PolyNet increases exploration without enforcing the first construction action, allowing its applicability to a wider range of CO problems

By utilizing a single decoder to learn multiple strategies, PolyNet quickly generates a set of diverse solutions for a problem instance. This significantly enhances exploration, allowing us to find better solution during training and testing. Furthermore, by abandoning the concept of forcing diverse first actions, we exclusively rely on PolyNet's inherent diversity mechanism to facilitate exploration during the search. This fundamental change improves solution generation for problems in which the first move greatly influences performance, such as in the CVRP with time windows (CVRPTW).

PolyNet is an autoregressive construction method, which makes a direct application to large-scale routing problems computationally impractical. However, small- and medium-scale routing problems, which often involve numerous constraints (e.g., customer time windows), are frequently encountered in practice. This is reflected in the current focus of the operations research community, which prioritizes smaller problems with numerous complex constraints (e.g., Zhang et al. (2024); Cataldo-Díaz et al. (2024); Lera-Romero et al. (2024); Jolfaei & Alinaghian (2024)). With PolyNet, we aim to take a step toward addressing these more complex problems by enabling diverse solution generation without relying on the symmetries characteristic of simpler routing problems.

We assess PolyNet's performance across four problems: the TSP, the CVRP, CVRPTW, and the flexible flow shop problem (FFSP). Across all problems, PolyNet consistently demonstrates a significant advancement over the state-of-the-art. Moreover, we observe that the solution diversity arising during PolyNet's training enables the discovery of superior solutions compared to artificially enforcing diversity by fixing the initial solution construction step.

## 2 LITERATURE REVIEW

**Neural CO** In their seminal work, Vinyals et al. (2015) introduce the pointer network architecture, an early application of modern machine learning methods to solve CO problems. Pointer networks autoregressively generate discrete outputs corresponding to input positions. When trained via supervised learning, they can produce solutions for the TSP with up to 50 nodes. Bello et al. (2016) propose training pointer networks using reinforcement learning instead and illustrate the efficacy of this method in solving larger TSP instances. However, these early methods are limited in generalization and in the quality of the generated solutions.

Nazari et al. (2018) extend the pointer network architecture to address the CVRP with up to 100 nodes. Building on this, Kool et al. (2019) enhance the architecture by introducing a transformer-based encoder with self-attention (Vaswani et al., 2017), enabling the model to learn more effective policies and produce higher-quality solutions. Recognizing that many CO problems exhibit inherent symmetries, Kwon et al. (2020) propose POMO, a method that exploits these symmetries to generate diverse solutions. However, POMO's effectiveness is heavily dependent on the assumption that such symmetries can be efficiently leveraged, which limits its applicability to simple problems. Kim et al. (2022) extend these ideas and propose a general-purpose symmetric learning scheme. Drakulic et al. (2023) use bisimulation quotienting (Givan et al., 2003) to improve out-of-distribution generalization of neural CO methods. However, they use imitation learning which requires a large training set of high-quality solutions limiting the applicability of their method. Luo et al. (2023) improves generalization in neural CO by using a "heavy" decoder to make more effective node selections

during solution construction, but this increases computational overhead. Despite these advancements, only a few works, such as Falkner & Schmidt-Thieme (2020), Kool et al. (2022a), Hua et al. (2025), and Berto et al. (2024b), target routing problems with additional constraints like time windows, highlighting a gap in extending neural CO approaches to more complex problem variants.

Instead of constructing solutions autoregressively, some methods predict a heat-map that emphasizes promising solution components (e.g., arcs in a graph) and is subsequently used in a post-hoc search for solution construction (Joshi et al., 2019; Fu et al., 2021; Kool et al., 2022b; Min et al., 2023). However, Xia et al. (2024) reveal that these methods often underperform compared to simpler baselines. Another class of methods iteratively improves initial solutions. Hottung & Tierney (2020), Ma et al. (2021), Chen & Tian (2019), and Hottung et al. (2025) iteratively improve an initial solution by performing local adjustments. Similarly, several works guide the $k$-opt heuristic for vehicle routing problems (Wu et al., 2019; da Costa et al., 2020; Ma et al., 2023). These improvement methods require longer runtimes to be effective. In contrast, PolyNet also excels at generating high-quality solutions quickly. Sun & Yang (2023) cast CO problems as discrete binary vector optimization problems and utilize denoising diffusion models for solution generation and Kim et al. (2024) utilize the solution symmetry in the combinatorial space to improve the sample efficiency during training. Both approaches fail to generalize to problems with more complex constraints like the CVRPTW.

Hottung et al. (2022) introduce efficient active search (EAS), a method that guides the search by updating a subset of the policy parameters during inference. Choo et al. (2022) propose SGBS, an inference mechanism that combines Monte-Carlo tree search with beam search to provide search guidance. When used in combination with EAS, it achieves state-of-the-art performance on several problems. Li et al. (2023b) learn a distribution of high-quality solutions using diffusion models and perform a gradient-based search at test time. PolyNet outperforms these recent approaches while being able to even tackle problems with more complex constraints.

**Diversity in metaheuristics** In the context of metaheuristics, solution diversity plays a crucial role in balancing the exploration-exploitation trade-off. Population-based algorithms, such as genetic algorithms, aim to maintain a pool of diverse solutions to promote thorough exploration of the search space (Gendreau et al., 2010). Additionally, some works specifically target generating multiple solutions for single-objective problems. For instance, Hanaka et al. (2022) introduce a polynomial-time algorithm to compute multiple diverse shortest paths in weighted directed graphs. Similarly, Huang et al. (2019) propose a niching memetic algorithm designed to identify multiple optimal solutions for TSP instances, emphasizing diversity within the solution set.

**Diversity mechanisms in RL** Diversity mechanisms in RL aim to encourage exploration by generating a range of strategies or solutions. Skill-learning algorithms, such as those by Eysenbach et al. (2018) and Sharma et al. (2019), aim to learn policies with diverse behaviors to accelerate training. However, these methods are not directly applicable to CO tasks, where diversity must also consider the quality of solutions, not just behavioral variation.

In contrast to implicitly maintaining a collection of agents through context, population-based RL techniques explicitly maintain a finite agent population and use diversity mechanisms to discover diverse strategies for solving RL tasks, e.g., Zhang et al. (2019), Pierrot & Flajolet (2023), Wu et al. (2023). In neural program synthesis, Bunel et al. (2018) optimize the expected reward when sampling a pool of solutions and keep the best one. Li et al. (2021) use the mutual information between agents' identities and trajectories as an intrinsic reward to promote diversity and thus solve cooperative tasks requiring diverse strategies.

**Diversity mechanisms in neural CO** Kim et al. (2021) present a hierarchical strategy for solving routing problems, where a base policy learns to generate diverse candidate solutions through entropy regularization. While entropy regularization can be used to increase diversity during solution sampling, it does not allow to learn truly different strategies. Xin et al. (2021) encourage diverse solutions using multiple decoders and KL divergence regularization, while Grinsztajn et al. (2023) use an agent population through multiple decoders to learn complementary strategies, updating exclusively the best-performing agent at each iteration similar to Bunel et al. (2018). While using multiple-decoders encourages diversity, it results in a large computational overhead. Hottung et al. (2021) and Chalumeau et al. (2023) learn a continuous latent space that encodes solutions for CO problems and use it to sample diverse solutions at test time. Both approaches are designed for longer runtimes and can not be easily applied to more difficult optimization problems.

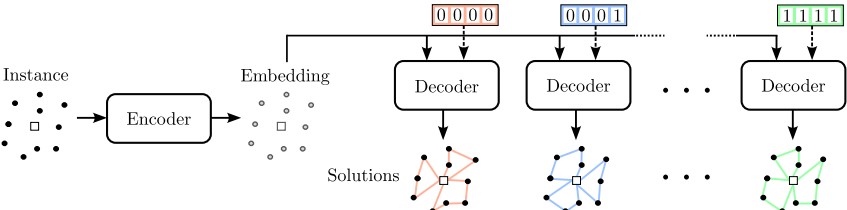

Figure 1: PolyNet solution generation.

## 3 POLYNET

### 3.1 BACKGROUND

Neural CO approaches seek to train a neural network denoted as $\pi_\theta$ with learnable weights $\theta$. The network's purpose is to generate a solution $\tau$ when provided with an instance $l$. To achieve this, we employ RL techniques and model the problem as a Markov decision process (MDP), wherein a solution is sequentially constructed in $T$ discrete time steps. At each step $t \in (1, \ldots, T)$, an action $a_t$ is selected based on the probability distribution $\pi_\theta(a_t|s_t)$ defined by the neural network where $s_t$ is the current state. The initial state $s_1$ encapsulates the information about the problem instance $l$, while subsequent states $s_{t+1}$ are derived by applying the action $a_t$ to the previous state $s_t$. A (partial) solution denoted as $\bar{\tau}_t$ is defined by the sequence of selected actions $(a_1, a_2, \ldots, a_t)$. Once a complete solution $\tau = \bar{\tau}_T$ satisfying all problem constraints is constructed, we can compute its associated reward $\mathcal{R}(\tau, l)$. The overall probability of generating a solution $\tau$ for an instance $l$ is defined as $\pi_\theta(\tau \mid l) = \prod_{t=1}^{T} \pi_\theta(a_t \mid s_t)$.

### 3.2 OVERVIEW

PolyNet is a learning-based approach designed to learn a set of diverse solution strategies for CO problems. During training, each strategy is allowed to specialize on a subset of the training data. This essentially results in a portfolio of strategies, which are known to be highly effective for solving CO problems (Bischl et al., 2016). Our pursuit of diversity is fundamentally a means to enhance exploration and consequently solution quality. Note that PolyNet not only enhances performance at test time (where we sample multiple solutions for each strategy and keep only the best one), but also improves exploration during training.

PolyNet aims to learn $K$ different solution strategies $\pi_1, \ldots, \pi_K$ using a single neural network. To achieve this, we condition the solution generation process on an additional input $v_i \in \{v_1, \ldots, v_K\}$ that defines which of the strategies should be used to sample a solution so that

$$\pi_1, \ldots, \pi_K = \pi_\theta(\tau_1 \mid l, v_1), \ldots, \pi_\theta(\tau_K \mid l, v_K). \tag{1}$$

We use a set of unique bit vectors for $\{v_1, \ldots, v_K\}$. Alternative representations should also be feasible as long as they are easily distinguishable by the network.

PolyNet uses a neural network that builds on the established transformer architecture for neural CO (Kool et al., 2019). The model consists of an encoder that creates an embedding of a problem instance, and a decoder that generates multiple solutions for an instance based on the embedding. To generate solutions quickly, we only insert the bit vector $v$ into the decoder, allowing us to generate multiple diverse solutions for an instance with only a single pass through the computationally expensive encoder. Figure 1 shows the overall solution generation process of the model where bit vectors of size 4 are used to generate to generate $K = 16$ different solutions for a CVRP instance.

### 3.3 TRAINING

During training we (repeatedly) sample $K$ solutions $\{\tau_1, \ldots, \tau_K\}$ for an instance $l$ based on $K$ different vectors $\{v_1, \ldots, v_K\}$, where the solution $\tau_i$ is sampled from the probability distribution $\pi_\theta(\tau_i \mid l, v_i)$. To allow the network to learn $K$ different solution strategies, we follow the Poppy method (Grinsztajn et al., 2023) and only update the model weights with respect to the best of the

$K$ solutions. Let $\tau^*$ be the best solution, i.e., $\tau^* = \arg\min_{\tau_i \in \{\tau_1,\ldots,\tau_K\}} \mathcal{R}(\tau_i, l)$, and let $v^*$ be the corresponding vector (ties are broken arbitrarily). We then update the model using the gradient

$$\nabla_\theta \mathcal{L} = \mathbb{E}_{\tau^*}\left[(R(\tau^*, l) - b_\circ)\nabla_\theta \log \pi_\theta(\tau^* \mid l, v^*)\right], \tag{2}$$

where $b_\circ$ is a baseline (which we discuss in detail below). Updating the model weights only based on the best found solution allows the network to learn specialized strategies that do not have to work well on all instances from the training set. While this approach does not explicitly enforce diversity, it incentivizes the model to learn diverse strategies in order to optimize overall performance. For a more in depth discussion on how this loss leads to diverse solution generation, see Grinsztajn et al. (2023).

**Exploration & baseline**   Many recent neural construction heuristics follow the POMO approach and rollout $N$ solutions from $N$ different starting states per instance to increase exploration. This is possible because many CO problems contain symmetries in the solution space that allow an optimal solution to be found from different states. In practice, this mechanism is implemented by forcing a different first construction action for each of the $N$ rollouts. Forcing diverse rollouts significantly improves exploration. However, this exploration mechanism should not be used when the first action can not be freely chosen without impacting the solution quality.

In PolyNet, we do not use an exploration mechanism that assumes symmetries in the solution space during training. Instead, we only rely on the exploration provided by our conditional solution generation. This allows us to solve a wider range of optimization problems. As a baseline we simply use the average reward of all $K$ rollouts for an instance, i.e., $b_\circ = \frac{1}{K}\sum_{i=1}^{K} R(\tau_i, l)$.

### 3.4   Network architecture

PolyNet extends the neural network architecture of the POMO approach by a new residual block in the decoder. This design allows us to start PolyNet's training from an already trained POMO model, which significantly reduces the amount of training time needed. Figure 2 shows the overall architecture of the modified decoder including the new PolyNet layers. The new layers accept the bit vector $v$ as input and use it to calculate an update for the output of the masked multi-head attention mechanism. They consist of a concatenation operation followed by two linear layers with a ReLU activation function in between. See Kwon et al. (2020) for more details on the encoder and decoder.

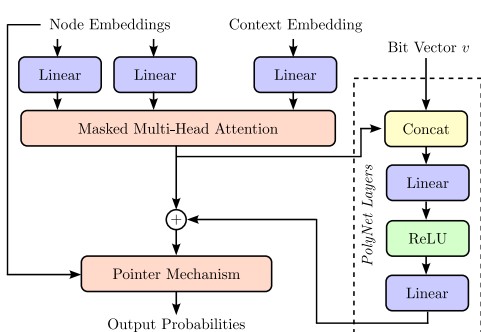

Figure 2: Decoder architecture.

The output of the new layers directly impacts the pointer mechanism used to calculate the output probabilities, allowing the new layers to significantly influence the solution generation based on $v$. However, the model can also learn to completely ignore $v$ by setting all weights of the second linear layer to zero. This is intentional, as our objective is to increase diversity via the loss function, rather than force unproductive diversity through the network architecture.

### 3.5   Search

A simple and fast search procedure given an unseen test instance $l$ is to sample multiple solutions in parallel and return the best one. Specifically, to construct a set of $M$ distinct solutions, we initially draw $M$ binary vectors from the set $v_1, \ldots, v_K$, allowing for replacement if $M$ exceeds the size of $K$. Subsequently, we employ each of these $M$ vectors to sample individual solutions. This approach generates a diverse set of instances in a parallel and independent manner, making it particularly suitable for real-world decision support settings where little time is available.

To facilitate a more extensive, guided search, PolyNet can be combined with EAS. EAS is a simple technique to fine-tune a subset of model parameters for a single instance in an iterative process driven by gradient descent. In contrast to the EAS variants described in Hottung et al. (2022), we do not insert any new layers into the network or update the instance embeddings. Instead, we only fine-tune the new PolyNet layers during search. Since PolyNet is specifically trained to create diverse solutions

based on these layers, EAS can easily explore a wide variety of solutions while modifying only a subset of the model parameters.

# 4 EXPERIMENTS

We compare PolyNet's search performance to state-of-the-art methods on four established problems. We also explore solution diversity during the training and testing phases and analyze the impact of the free first move selection. An ablation study for our network architecture changes is provided in Appendix B. While most of our evaluation is focused on routing problems, we demonstrate that PolyNet can also be used to solve other CO problems by applying PolyNet to the FFSP (see Appendix A). Our experiments are conducted on a GPU cluster utilizing a single Nvidia A100 GPU per run. In all experiments, the new PolyNet layers comprise two linear layers, each with a dimensionality of 256. Our implementation of PolyNet is available at `https://github.com/ahottung/PolyNet`. Furthermore, PolyNet is also implemented in the RL4CO framework (Berto et al., 2024a).

## 4.1 PROBLEMS

**TSP**   The TSP is a thoroughly researched routing problem in which the goal is to find the shortest tour among a set of $n$ nodes. The tour must visit each node exactly once and then return to the initial starting node. We consider the TSP due to its significant attention from the machine learning (ML) community, making it a well-established benchmark for ML-based optimization approaches. However, it is important to note that instances with $n \leq 300$ can be quickly solved to optimality by CO solvers that have been available for many years. To generate problem instances, we adhere to the methodology outlined in Kool et al. (2019).

**CVRP**   The objective of the CVRP is to determine the shortest routes for a fleet of vehicles tasked with delivering goods to a set of $n$ customers. These vehicles start and conclude their routes at a depot and have a limited capacity of goods that they can carry. CVRP instances are considerably more difficult to solve than TSP instances of equivalent size (despite both problems being of the same computational complexity class). Even cutting-edge CO methods struggle to reliably find optimal solutions for instances with $n \leq 300$ customers. To generate problem instances, we again adopt the approach outlined in Kool et al. (2019).

**CVRPTW**   The CVRPTW is an extension of the CVRP that introduces temporal constraints limiting when a customer can receive deliveries from a vehicle. Each customer $i$ is associated with a time window, comprising an earliest arrival time $e_i$ and a latest allowable arrival time $l_i$. Vehicles may arrive early at a customer $i$, but they must wait until the specified earliest arrival time $e_i$ before making a delivery. The travel duration between customer $i$ and customer $j$ is calculated as the Euclidean distance between their locations, and each delivery has a fixed duration. All vehicles initiate their routes from the depot at time $0$. We use the CVRP instance generator outlined in Queiroga et al. (2022) to generate customer positions and demands, and we adhere to the methodology established in Solomon (1987) for generating the time windows. It's essential to note that customer positions are not sampled from a uniform distribution; instead, they are clustered to replicate real-world scenarios. Further details for instance generation are given in Appendix F.

**FFSP**   The description of the FFSP and all experimental results can be found Appendix A.

## 4.2 SEARCH PERFORMANCE

We conduct an extensive evaluation of PolyNet's search performance, benchmarking it against state-of-the-art methods. For the considered routing problems, we train separate models for problem instances of size 100 and 200, and then evaluate the models trained on $n = 100$ using instances with 100 and 150 nodes, and the models trained on $n = 200$ using instances with 200 and 300 nodes. We can thus assess the model's capability to generalize to instances that diverge from the training data. Throughout our evaluation, we employ the instance augmentation technique of Kwon et al. (2020).

Table 1: Search performance results for TSP.

| | Method | **Test** (10K instances) $n_{tr} = n_{eval} = 100$ | | | **Test** (1K instances) $n_{tr} = n_{eval} = 200$ | | | **Generalization** (1K instances) $n_{tr} = 100, n_{eval} = 150$ | | | $n_{tr} = 200, n_{eval} = 300$ | | |
|---|---|---|---|---|---|---|---|---|---|---|---|---|---|
| | | Obj. | Gap | Time | Obj. | Gap | Time | Obj. | Gap | Time | Obj. | Gap | Time |
| | Concorde | 7.765 | - | 82m | 10.687 | - | 31m | 9.346 | - | 17m | 12.949 | - | 83m |
| | LKH3 | 7.765 | 0.000% | 8h | 10.687 | 0.000% | 3h | 9.346 | 0.000% | 99m | 12.949 | 0.000% | 5h |
| Unguided | POMO greedy | 7.775 | 0.135% | 1m | 10.770 | 0.780% | <1m | 9.393 | 0.494% | <1m | 13.216 | 2.061% | 1m |
| | sampling | 7.772 | 0.100% | 3m | 10.759 | 0.674% | 2m | 9.385 | 0.411% | 1m | 13.257 | 2.378% | 7m |
| | Poppy | 7.766 | 0.015% | 4m | 10.711 | 0.226% | 2m | 9.362 | 0.164% | 1m | 13.052 | 0.793% | 7m |
| | PolyNet | 7.765 | **0.000%** | 4m | 10.690 | **0.032%** | 2m | 9.352 | **0.055%** | 1m | 12.995 | **0.351%** | 8m |
| Guided | DPDP | 7.765 | 0.004% | 2h | - | - | - | 9.434 | 0.937% | 44m | - | - | - |
| | COMPASS | 7.765 | 0.002% | 2h | - | - | - | 9.354 | 0.083% | 32m | - | - | - |
| | MDAM | 7.781 | 0.208% | 4h | - | - | - | 9.403 | 0.603% | 1h | - | - | - |
| | POMO EAS | 7.769 | 0.053% | 3h | 10.720 | 0.310% | 3h | 9.363 | 0.172% | 1h | 13.048 | 0.761% | 8h |
| | SGBS | 7.769 | 0.058% | 9m | 10.727 | 0.380% | 24m | 9.367 | 0.220% | 8m | 13.073 | 0.951% | 77m |
| | SGBS+EAS | 7.767 | 0.035% | 3h | 10.719 | 0.300% | 3h | 9.359 | 0.136% | 1h | 13.050 | 0.776% | 8h |
| | PolyNet EAS | 7.765 | **0.000%** | 3h | 10.687 | **0.001%** | 2h | 9.347 | **0.001%** | 1h | 12.952 | **0.018%** | 7h |

For the training of PolyNet models, we set the parameter $K$ to 128 across all problems. We use a learning rate of $10^{-4}$ for the TSP and CVRP and $10^{-5}$ for the CVRPTW. For instances of size $n=100$, we train our models for 300 epochs (200 for the TSP), with each epoch comprising $4 \times 10^8$ solution rollouts. Note that we warm-start the training for these models using previously trained POMO models to reduce training times (results for cold-starting training can be found in Appendix C). For instances with $n=200$, we start training based on the $n=100$ PolyNet models, running 40 additional training epochs (20 for the TSP). To optimize GPU memory utilization, we adjust the batch size separately for each problem and its dimensions.

We categorize the evaluated algorithms into two groups: unguided and guided methods. Unguided algorithms generate solutions independently, while guided methods incorporate a high-level search component capable of navigating the search space. For a comparison to unguided algorithms, we compare PolyNet to POMO and the Poppy approach with a population size of 8. To ensure fairness we retrain Poppy using the same training setup as for PolyNet. Note that POMO has already been trained to full convergence and does not benefit from additional training (see Figure 3). For all approaches, we sample $64 \times n$ solutions per instance (except for POMO using greedy solution generation). For our comparison to guided algorithms, we use PolyNet with EAS and compare it with POMO combined with EAS (Hottung et al., 2022) and SGBS (Choo et al., 2022). For PolyNet, we sample $200 \times 8 \times n$ solutions per instance over the course of 200 iterations. Furthermore, we compare to some problem-specific approaches that are explained below. Note that we provide additional search trajectory plots in Appendix E. Results for the FFSP can be found in Appendix A.

**TSP** We use the 10,000 test instances with $n=100$ from Kool et al. (2019) and test sets consisting of $1,000$ instances from Hottung et al. (2021) for $n=150$ and $n=200$. For $n=300$, we generate new instances. As a baseline, we use the exact solver Concorde (Applegate et al., 2006) and the heuristic solver LKH3 (Helsgaun, 2017). Additionally, we also compare to DPDP (Kool et al., 2022b), COMPASS (Chalumeau et al., 2023), and the diversity-focused method MDAM (Xin et al., 2021) with a beam search width of 256.

Table 1 provides our results on the TSP, showing clear performance improvements of PolyNet during fast solution generation and extensive search with EAS for all considered instance sets. For TSP instances with 100 nodes, PolyNet achieves a gap that is practically zero while being roughly 120 times faster than LKH3. Furthermore, on all four instance sets, PolyNet with unguided solution sampling finds solutions with significantly lower costs in comparison to guided learning approaches while reducing the runtime by a factor of more than 100 in many cases.

**CVRP** Similar to the TSP, we use the test sets from Kool et al. (2019) and Hottung et al. (2021). As a baseline, we use LKH3 (Helsgaun, 2000) and the state-of-the-art (OR) solver HGS (Vidal et al., 2012; Vidal, 2022). We compare to the same learning methods as for the TSP with the addition of DACT (Ma et al., 2021).

Table 2: Search performance results for CVRP.

| | Method | Test (10K instances) $n_{tr}=n_{eval}=100$ | | | Test (1K instances) $n_{tr}=n_{eval}=200$ | | | Generalization (1K instances) $n_{tr}=100, n_{eval}=150$ | | | $n_{tr}=200, n_{eval}=300$ | | |
|---|---|---|---|---|---|---|---|---|---|---|---|---|---|
| | | Obj. | Gap | Time | Obj. | Gap | Time | Obj. | Gap | Time | Obj. | Gap | Time |
| | HGS | 15.563 | - | 54h | 21.766 | - | 17h | 19.055 | - | 9h | 27.737 | - | 46h |
| | LKH3 | 15.646 | 0.53% | 6d | 22.003 | 1.09% | 25h | 19.222 | 0.88% | 20h | 28.157 | 1.51% | 34h |
| Unguided | POMO greedy | 15.754 | 1.23% | 1m | 22.194 | 1.97% | <1m | 19.684 | 3.30% | <1m | 28.627 | 3.21% | 1m |
| | sampling | 15.705 | 0.91% | 5m | 22.136 | 1.70% | 3m | 20.109 | 5.53% | 1m | 28.613 | 3.16% | 9m |
| | Poppy | 15.685 | 0.78% | 5m | 22.040 | 1.26% | 3m | 19.578 | 2.74% | 1m | 28.648 | 3.28% | 8m |
| | PolyNet | 15.640 | **0.49%** | 5m | 21.957 | **0.88%** | 3m | 19.501 | **2.34%** | 1m | 28.552 | **2.94%** | 8m |
| Guided | DACT | 15.747 | 1.18% | 22h | - | - | - | 19.594 | 2.83% | 16h | - | - | - |
| | DPDP | 15.627 | 0.41% | 23h | - | - | - | 19.312 | 1.35% | 5h | - | - | - |
| | COMPASS | 15.594 | 0.20% | 4h | - | - | - | 19.313 | 1.35% | 1h | - | - | - |
| | MDAM | 15.885 | 2.07% | 5h | - | - | - | 19.686 | 3.31% | 1h | - | - | - |
| | POMO EAS | 15.618 | 0.35% | 6h | 21.900 | 0.61% | 3h | 19.205 | 0.79% | 2h | 28.053 | 1.14% | 12h |
| | SGBS | 15.659 | 0.62% | 10m | 22.016 | 1.15% | 7m | 19.426 | 1.95% | 4h | 28.293 | 2.00% | 22m |
| | SGBS+EAS | 15.594 | 0.20% | 6h | 21.866 | 0.46% | 4h | 19.168 | 0.60% | 2h | 28.015 | 1.00% | 12h |
| | PolyNet EAS | 15.584 | **0.14%** | 4h | 21.821 | **0.25%** | 2h | 19.166 | **0.59%** | 1h | 27.993 | **0.92%** | 9h |

Table 3: Search performance results for CVRPTW.

| | Method | Test (10K instances) $n_{tr}=n_{eval}=100$ | | | Test (1K instances) $n_{tr}=n_{eval}=200$ | | | Generalization (1K instances) $n_{tr}=100, n_{eval}=150$ | | | $n_{tr}=200, n_{eval}=300$ | | |
|---|---|---|---|---|---|---|---|---|---|---|---|---|---|
| | | Obj. | Gap | Time | Obj. | Gap | Time | Obj. | Gap | Time | Obj. | Gap | Time |
| | PyVRP | 12,534 | - | 39h | 18,422 | - | 11h | 17,408 | - | 9h | 25,732 | - | 26h |
| Unguided | POMO greedy | 13,120 | 4.67% | 1m | 19,656 | 6.70% | 1m | 18,670 | 7.25% | <1m | 28,022 | 8.90% | 2m |
| | sampling | 13,019 | 3.87% | 7m | 19,531 | 6.02% | 4m | 18,571 | 6.68% | 2m | 28,017 | 8.88% | 13m |
| | Poppy | 12,969 | 3.47% | 5m | 19,406 | 5.34% | 3m | 18,612 | 6.91% | 2m | 28,104 | 9.22% | 10m |
| | PolyNet | 12,876 | **2.73%** | 5m | 19,232 | **4.40%** | 3m | 18,429 | **5.86%** | 2m | 27,807 | **8.07%** | 10m |
| Guided | POMO EAS | 12,762 | 1.81% | 6h | 18,966 | 2.96% | 4h | 17,851 | 2.54% | 2h | 26,608 | 3.40% | 14h |
| | SGBS | 12,897 | 2.89% | 12m | 19,240 | 4.44% | 8m | 18,201 | 4.55% | 4m | 27,264 | 5.95% | 25m |
| | SGBS+EAS | 12,714 | 1.43% | 7h | 18,912 | 2.66% | 4h | 17,835 | 2.45% | 2h | 26,651 | 3.57% | 15h |
| | PolyNet EAS | 12,654 | **0.96%** | 5h | 18,739 | **1.72%** | 3h | 17701 | **1.68%** | 1h | 26,504 | **3.00%** | 10h |

The CVRP results in Table 2 once again indicate consistent improvement across all considered problem sizes. Especially on the instances with 100 and 200 customers, PolyNet improves upon the state-of-the-art learning-based approaches by reducing the gap by more than 30% during fast solution generation and extensive search. Also note that PolyNet significantly outperforms the other diversity-focused approaches Poppy and MDAM.

**CVRPTW** For the CVRPTW, we use the state-of-the-art CO solver PyVRP (Version 0.5.0) (Wouda et al., 2024) as a baseline stopping the search after 1,000 iterations without improvement. We compare to a POMO implementation that we adjusted to solve the CVRPTW by extending the node features with the time windows and the context information used at each decoding step by the current time point. These models are trained for 50,000 epochs, mirroring the training setup used for the CVRP.

Table 3 presents the CVRPTW results, demonstrating PolyNet's consistent and superior performance across all settings compared to Poppy and POMO (with SGBS and EAS). Notably, for instances with 100 customers, PolyNet matches almost the CO solver PyVRP with a gap below 1%.

## 4.3 DIVERSITY

**Does PolyNet's training encourage diversity?** To assess the effectiveness of our diversity mechanism, we conduct short training runs of PolyNet for the TSP, CVRP, and CVRPTW with varying values of $K$. As a baseline we report results for the training of the POMO model. For all runs, we use a batch size of $480$ and a learning rate set at $10^{-4}$. To ensure a stable initial state for training runs, we start all runs from a trained POMO model. For PolyNet, we initialize the PolyNet layer weights to zero, minimizing initial randomness. During training, we perform regular evaluations on a separate validation set of $10\,000$ instances, sampling $800$ solutions per instance.

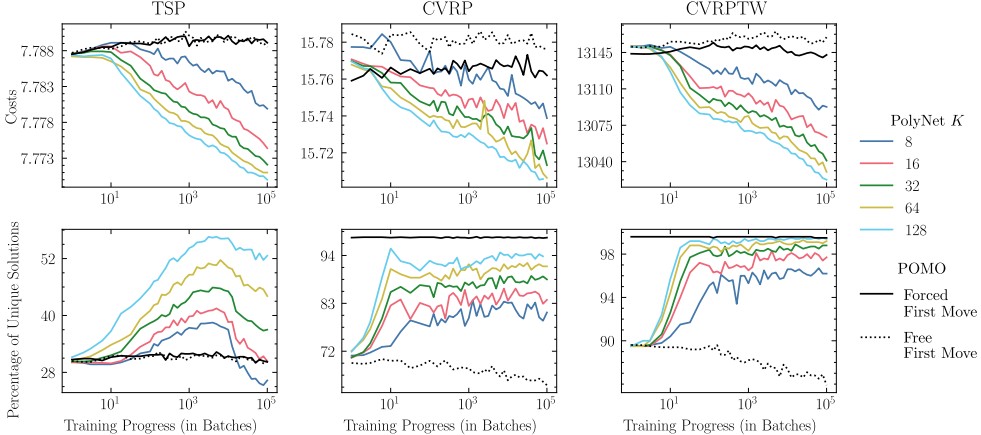

Figure 3: Validation performance during training (log scale).

Table 4: Solution diversity measured using avg. broken pairs distance.

| Method | TSP | CVRP | CVRPTW |
|---|---|---|---|
| PolyNet | 4.621 | 18.535 | 19.969 |
| POMO | 2.400 | 16.785 | 18.162 |

Table 5: Ablation results for free first move selection.

| Method | | TSP Gap | Time | CVRP Gap | Time | CVRPTW Gap | Time |
|---|---|---|---|---|---|---|---|
| PolyNet    Free first move | | 0.000% | 4m | 0.49% | 5m | 2.73% | 5m |
| Forced first move | | 0.006% | 4m | 0.59% | 5m | 3.00% | 6m |
| Poppy | | 0.015% | 4m | 0.78% | 5m | 3.47% | 5m |

Our evaluation tracks the cost of the best solution and the percentage of unique solutions among the 800 solutions per instance. Figure 3 presents the evaluation results. Across all three problems, we observe a clear trend: higher values of $K$ lead to a more rapid reduction in average costs on the validation set and are associated with a greater percentage of unique solutions. Note that POMO does not benefit from further training. These results underscore the effectiveness of our approach in promoting solution diversity and improving solution quality.

**Does PolyNet generate diverse solutions at test time?** In this experiment we evaluate the diversity of PolyNet at test time. As a baseline we use the POMO approach, which enforces diverse solution generation by forcing the first action. For each test instance, we sample 100 solutions with both approaches and calculate the average broken pairs distance (Prins, 2009), which is an established method to measure diversity for routing problem solutions (see, e.g., Vidal et al. (2012)). The broken pairs distance $d(A, B)$ for two solutions $A$ and $B$ is the number of edges of $A$ that are not in $B$. We measure the diversity of a set of solutions using the average across the broken pairs distances between all solution pairs of the set. The results shown in Table 4 confirm that PolyNet is able to generate more diverse solutions than POMO at test time. An additional visual comparison of the differences in solution diversity between POMO and PolyNet can be found in Appendix D.

**Does PolyNet encourage *useful* diversity?** As discussed in the introduction, solution diversity alone is neither inherently beneficial nor difficult to achieve (e.g., by generating random solutions). What truly matters is that the diversity is *useful*, meaning it contributes to finding high-quality solutions. To assess the diversity of PolyNet we conduct two separate experiments.

First, we evaluate the importance of each of the $K$ learned strategies at test time by counting how often each strategy finds the best solution across 10 000 test instances. The results, shown in Figure 4, indicate that all $K$ strategies contribute significantly to performance. Even the least effective strategies identify the best solution on 544, 61, and 54 instances for the TSP, CVRP, and CVRPTW, respectively. Note that for the TSP, multiple strategies often converge to the same best solution, likely because most instances are solved to optimality. However, for the CVRP and CVRPTW, the best solution for an instance is typically found by only one strategy. These results demonstrate that all learned strategies are able to generate high-quality solutions and that all strategies contribute meaningfully to the overall sampling performance of PolyNet.

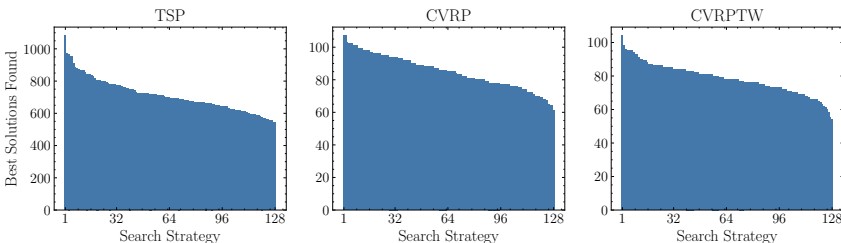

Figure 4: Contribution of different search strategies. Strategies are sorted based on their contribution.

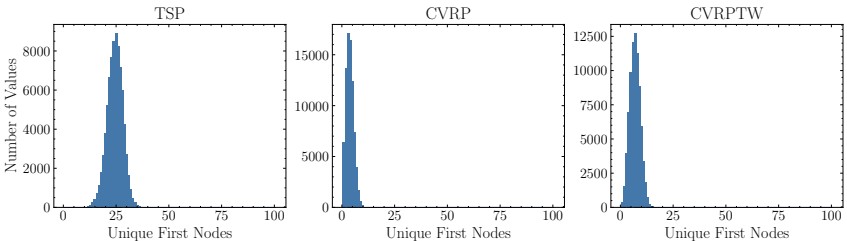

Figure 5: Frequency plots for the number of unique first nodes.

Second, we evaluate the number of distinct first nodes selected by the $K$ different strategies of PolyNet. This provides insights into the diversity achieved by PolyNet, particularly in comparison to POMO, which enforces diversity by selecting different first actions. As shown in Figure 5, PolyNet selects fewer distinct first nodes than POMO. On average, PolyNet selects 24.5, 3.8, and 7.0 distinct first nodes for the TSP, CVRP, and CVRPTW, respectively. Despite selecting fewer different first nodes, PolyNet is able to generate solutions that are more diverse overall than those generated by POMO, as shown in Table 4. This suggests that PolyNet achieves diversity through means other than enforcing the first action.

### 4.4 ABLATION STUDY: FORCING THE FIRST MOVE

PolyNet does not force diverse first construction actions and relies solely on its built-in diversity mechanism to select the first node. To assess this approach, we compare the performance of PolyNet with and without forced first move selection when sampling $64 \times 100$ solutions per instance.

Table 5 shows the results for all problems with $n = 100$. Remarkably, across all scenarios, allowing PolyNet to select the first move yields superior performance compared to forcing the first move. Furthermore, PolyNet with forced first move selection outperforms Poppy (which also enforces the first move), underscoring that PolyNet's single-decoder architecture delivers better results.

## 5 CONCLUSION

We introduced the novel approach PolyNet, which is capable of learning diverse solution strategies using a single-decoder model. PolyNet deviates from the prevailing trend in neural construction methods, in which diverse first construction steps are forced to improve exploration. Instead, it relies on its diverse strategies for exploration, enabling its seamless adaptation to problems where the first move significantly impacts solution quality. In our comprehensive evaluation across four problems, including the more challenging CVRPTW, PolyNet consistently demonstrates performance improvements over all other learning-based methods, particularly those focused on diversity.

Regarding our approach's limitations, we acknowledge that the computational complexity of the attention mechanism we employ restricts its applicability to instances with less than 1000 nodes. However, it is essential to emphasize that the problem sizes examined in this paper for the CVRP(TW) remain challenging for traditional CO solvers and are highly significant in real-world applications. Furthermore, we note that the black-box nature of the PolyNet's decision-making may be unacceptable in certain decision contexts.

ACKNOWLEDGEMENTS

André Hottung was supported by the Deutsche Forschungsgemeinschaft (DFG, German Research Foundation) under Grant No. 521243122. Additionally, we gratefully acknowledge the funding of this project by computing time provided by the Paderborn Center for Parallel Computing (PC2). Furthermore, some computational experiments in this work have been performed using the Bielefeld GPU Cluster. We thank the HPC.NRW team for their support.

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

## A    EXPERIMENTAL RESULTS FOR THE FFSP

We examine the flexible flow shop problem (FFSP) to further demonstrate PolyNet's versatility beyond routing problems, and the generality of the technique beyond the POMO architecture. In the FFSP, a set of $n$ jobs must be processed in a series of stages, with each stage comprising multiple machines that each require different processing times. A machine cannot process more than one job at the same time. The objective is to determine a job schedule with the shortest makespan. We generate problem instances randomly according to the method in Kwon et al. (2021). We experiment on FFSP instances with $n = 20$, 50, and 100 jobs, where each instance consists of 3 stages and each stage has 4 machines.

**Network architecture**    We use a neural network that extends the architecture of MatNet (Kwon et al., 2021). More concretely, we introduce a residual block for each of the decoding networks. In all experiments, the PolyNet layers in every block comprise two linear layers, each with a dimensionality of 256.

**Training**    To train the PolyNet models, we use a learning rate set to $10^{-4}$, and batch sizes of 256, 128, and 32 for instances comprising $n = 20$, 50, and 100 jobs, respectively. We set the parameter $K$ to 24 across all problem sizes. The training runs start from trained MatNet models and continue for 100 (20 for $n = 100$) epochs, with each epoch comprising $24 \times 10^5$ solution rollouts.

**Search performance**    For the FFSP, we use the randomly generated test sets from Kwon et al. (2021) and comparison algorithms. In addition to MatNet (Kwon et al., 2021), we compare our method to CPLEX (CPLEX-Optimization-Studio, 2020) with mixed-integer programming models and metaheuristic solvers. Table 6 provides the results for the FFSP. The gaps are reported with respect to PolyNet with $\times128$ augmentation result, as optimal solutions are not available. The results demonstrate PolyNet's consistent improvement in performance across all problem sizes. Notably, PolyNet outperforms MatNet-based solvers for the FFSP on all problem sizes and produces solutions of even higher quality within similar run times. Given the different architecture of MatNet compared to those used for routing problems, the results highlight that the performance gains from PolyNet are not confined to a specific network architecture. Moreover, these gains extend beyond routing problems, indicating PolyNet's broad applicability across various CO problems.

Table 6: Search performance results for FFSP.

| Method | FFSP20 | | | FFSP50 | | | FFSP100 | | |
|---|---|---|---|---|---|---|---|---|---|
| | MS | Gap | Time | MS | Gap | Time | MS | Gap | Time |
| CPLEX (60s) | 46.4 | 21.5 | (17h) | $\times$ | | | $\times$ | | |
| CPLEX (600s) | 36.6 | 11.7 | (167h) | | | | | | |
| Random | 47.8 | 22.9 | (1m) | 93.2 | 44 | (2m) | 167.2 | 78.0 | (3m) |
| Shortest Job First | 31.3 | 6.4 | (40s) | 57.0 | 7.8 | (1m) | 99.3 | 10.1 | (2m) |
| Genetic Algorithm | 30.6 | 5.7 | (7h) | 56.4 | 7.2 | (16h) | 98.7 | 9.5 | (29h) |
| Particle Swarm Opt. | 29.1 | 4.2 | (13h) | 55.1 | 5.9 | (26h) | 97.3 | 8.1 | (48h) |
| MatNet | 27.1 | 2.2 | (5s) | 51.5 | 2.3 | (9s) | 91.6 | 2.4 | (17s) |
| PolyNet | 26.7 | 1.8 | (5s) | 51.0 | 1.8 | (10s) | 91.2 | 2.0 | (18s) |
| MatNet ($\times128$) | 25.4 | 0.5 | (3m) | 49.6 | 0.4 | (8m) | 89.8 | 0.6 | (19m) |
| PolyNet ($\times128$) | 24.9 | - | (4m) | 49.2 | - | (9m) | 89.2 | - | (23m) |

## B    ABLATION STUDY: POLYNET LAYERS

To assess the influence of our new PolyNet layers, we perform an ablation experiment in which we systematically remove these additional layers during the training. In this modified version of PolyNet, we add the vector $v$ directly to the output of the masked multi-head attention mechanism (shown in Figure 2). To achieve this, the vector $v$ is zero-padded to match the necessary dimensions. Note that this altered PolyNet configuration has exactly the same number of total model parameters as the POMO model.

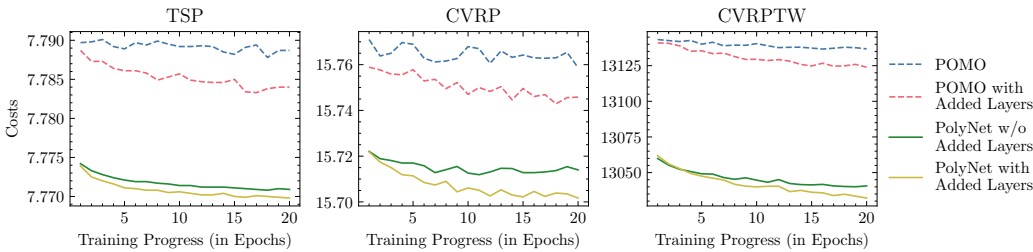

Figure 6: Validation performance during training.

For a fair comparison we also extend the baseline POMO model by our additional PolyNet layers. The modified POMO model comprises the same number of model parameters as our PolyNet architecture. Note that the modified POMO model does not accept a vector $v$. Instead, the PolyNet layers directly receive the output of the masked multi-head attention mechanism.

We train all models for 20 epochs using the same training hyperparameters as for our main training run described in Section 4.2. After each epoch we evaluate the model performance on our validation set that comprises $10,000$ instances, sampling 800 solutions per instance.

Figure 6 shows the validation costs over the course of the training. On all three problems, PolyNet without the added layers performs worse than its original version, which demonstrates the importance of the PolyNet layers. Nonetheless, even PolyNet without the added layers is able to significantly outperform POMO with additional layers.

## C  POLYNET TRAINING: WARM-START VS. COLD-START

In all of our previous experiments, we initialize the training process using a pre-trained POMO model. To assess the implications of this setup, we conduct additional training runs where models are either warm-started or trained from scratch. These runs utilize identical hyperparameters as our primary training sessions, but extend training for a total of 400 epochs. It's worth noting that these extended training runs span several weeks and that the additional insights gained by even longer training runs are likely limited.

Figure 7 illustrates the validation performance throughout these training sessions. We observe notable benefits from warm-starting the training process. Specifically, on the CVRP and CVRPTW tasks, models initialized from a cold start consistently exhibit poorer performance even after 400 epochs of training. This discrepancy is particularly evident in the CVRPTW task, where the lower learning rate of $10^{-5}$ leads to slower training progress. Only in the TSP task does the performance of the cold-started model converge with that of the warm-started model after 400 epochs. We thus infer that while PolyNet can indeed be trained from scratch, warm-starting the training process is advisable to mitigate computational costs.

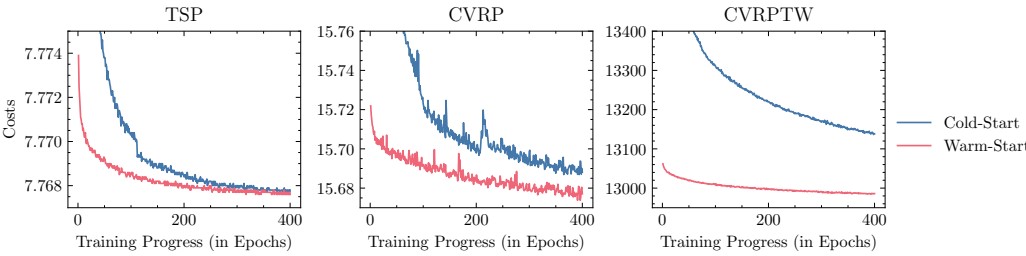

Figure 7: Validation performance during training: Starting training from scratch (cold-start) vs. starting from a trained POMO model (warm-start).

# D    SOLUTION UNIQUENESS VS. COSTS PLOTS

In this experiment, we compare the solutions generated by POMO to those generated by PolyNet with respect to their cost and their uniqueness on an instance-by-instance basis. More specifically, for a subset of test instances, we sample 100 solutions per instance with both approaches and compare the solutions found on the basis of their uniqueness and their cost. We calculate the uniqueness of a solution by using the average of the broken pairs distance (Prins, 2009) to all other 99 generated solutions.

Figures 8-10 shows the results for the first six TSP, CVRP, and CVRPTW test instances. PolyNet generates more unique solutions on most instances. Especially on the TSP the solutions from PolyNet are much more diverse, however this diversity occasionally comes at the expense of solution quality. Since we ultimately only care about the solution with the minimal costs in the generated solution set and, not the average solution quality, this is not a limitation of the method. PolyNet is only able to increase its performance in the minimum cost case by sometimes trying out solutions that a greedier method, like POMO, ignores.

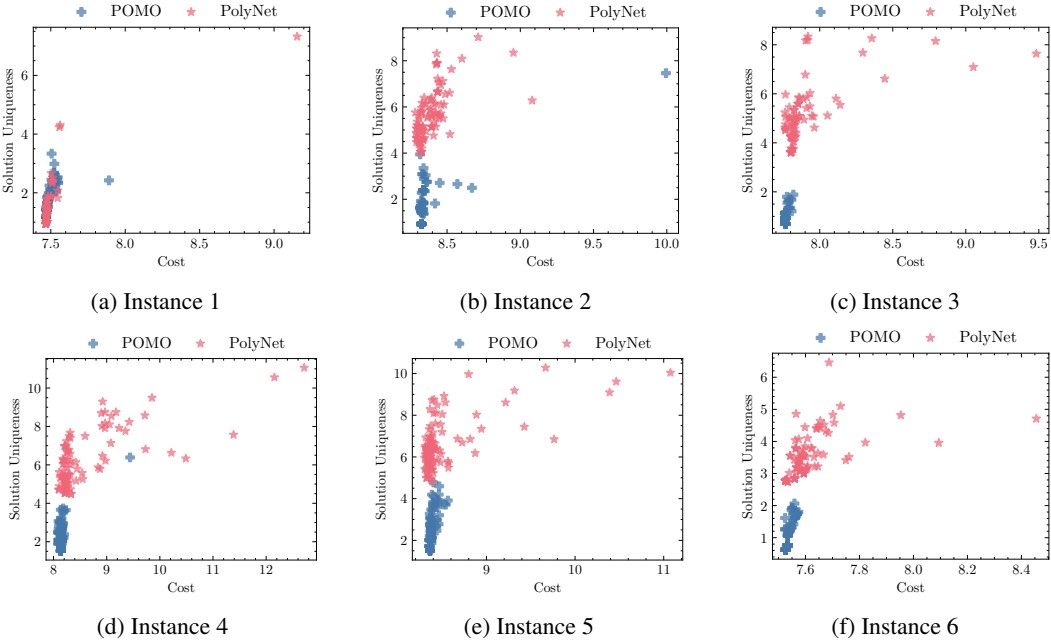

Figure 8: Solution diversity vs. costs for the TSP.

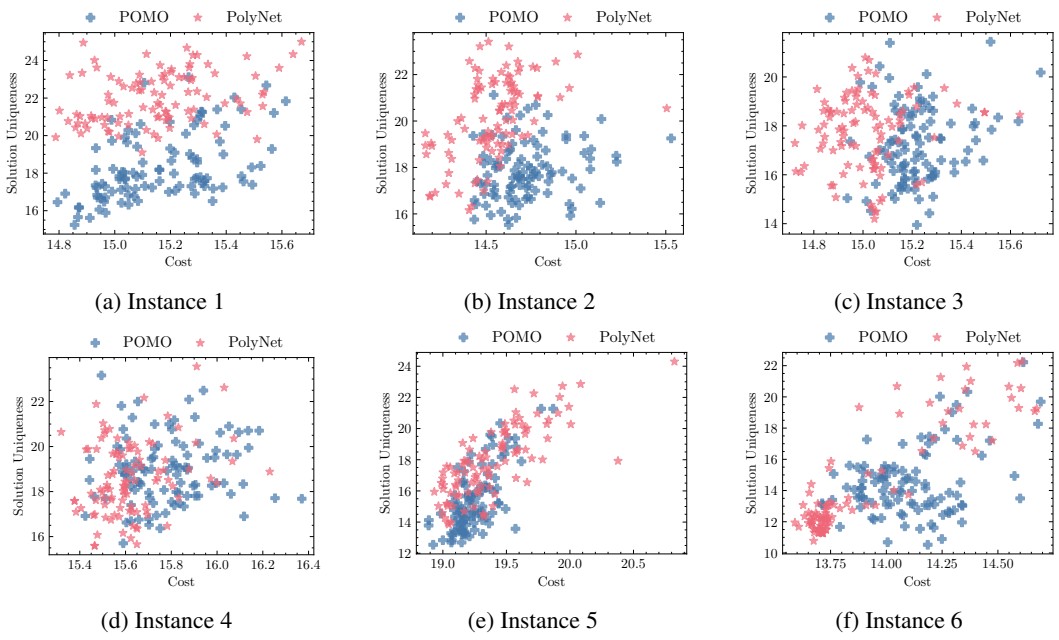

Figure 9: Solution diversity vs. costs for the CVRP

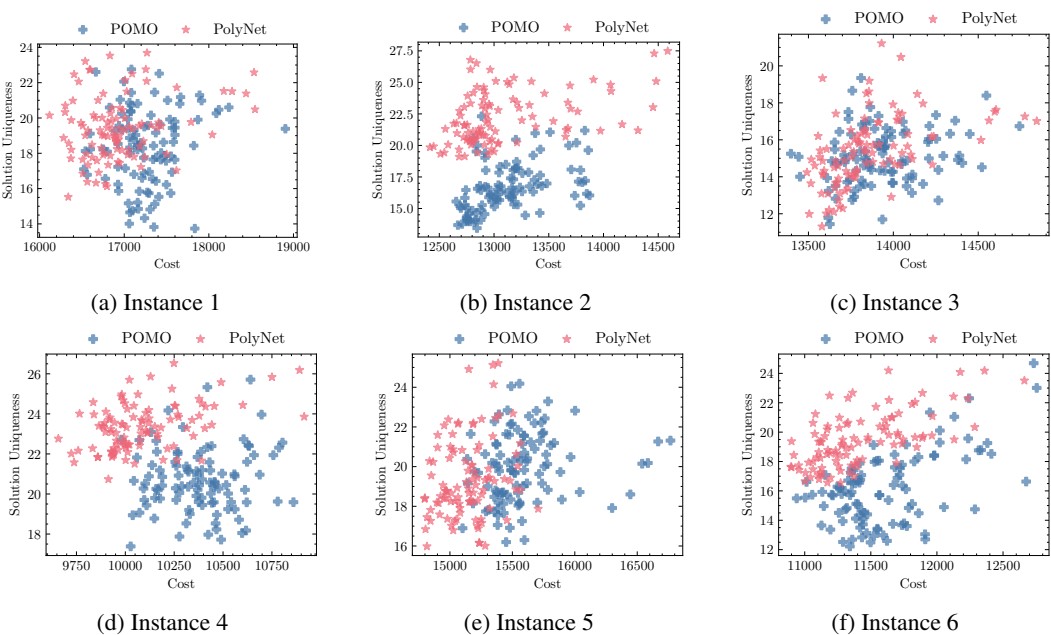

Figure 10: Solution diversity vs. costs for the CVRPTW

# E  SEARCH TRAJECTORY ANALYSIS

In Figure 11, we show the search trajectories for models trained with varying values of $K$ across all three routing problems featuring 100 nodes. The search process employs solution sampling without EAS and without the use of instance augmentations. These models used for the search undergo training for 150 epochs (except for the CVRP, where the training spans 200 epochs).

It is evident across all three problems that the search does not achieve full convergence within 10,000 iterations. This observation once again underscores PolyNet's capability to discover diverse solutions, enabling it to yield improved results with extended search budgets. It's important to note that this experiment, including model training, has not been replicated with multiple seeds. Nevertheless, the results suggest that models trained with larger $K$ values benefit more from longer search budgets compared to models trained with smaller values.

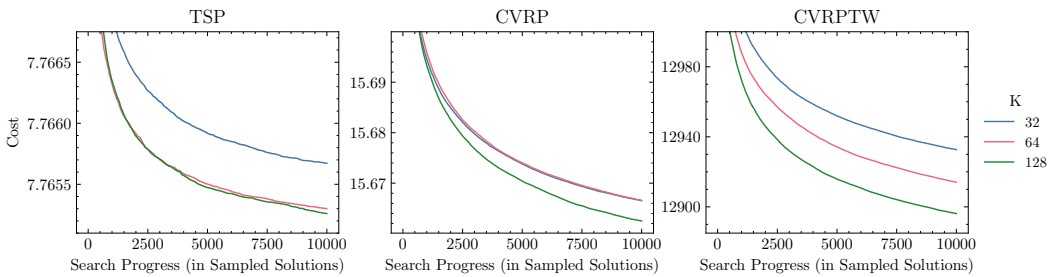

Figure 11: Search trajectories.

# F  CVRPTW INSTANCE GENERATION

We generate instances for the CVRPTW with the goal of including real-world structures. To achieve this, we employ a two-step approach. First, we use the CVRP instance generator developed by Queiroga et al. (2022) to produce the positions and demands of customers. Subsequently, we follow the methodology outlined by Solomon (1987) to create the time windows. We generate different instance sets for training, validation, and testing.

Customer positions are generated using the *clustered* setting (configuration 2) and customer demands are based on the *small, large variance* setting (configuration 2). The depot is always *centered* (configuration 2). It is worth noting that the instance generator samples customer positions within the 2D space defined by $[0, 999]^2$. Independently from the instance generator, vehicle capacities are set at 50 for instances involving fewer than 200 customers and increased to 70 for instances with 200 or more customers.

To generate the time windows, we adhere to the procedure outlined by Solomon (1987) for instances with randomly clustered customers (i.e., we do not utilize the 3-opt technique to create reference routes). We randomly generate time windows $(e_i, l_i)$ for all customers, and set 2400 as the latest possible time for a vehicle to return to the depot. The time window generation process, as described in Solomon (1987), limits the time windows to ensure feasibility (e.g., by selecting $l_i$ so that there is always sufficient time for servicing the customer and returning to the depot). The center of the time window is uniformly sampled from range defined by these limits. We set the maximum width of the time window to 500 and the service duration to 50. These parameter values have been deliberately chosen to strike a balance between the constraints of vehicle capacity and time windows, requiring both aspects to be considered during the solution generation proces.

