# OpenReview forum: "PolyNet: Learning Diverse Solution Strategies for Neural Combinatorial Optimization"
_ICLR.cc/2025/Conference — ICLR 2025 Poster_

### Official Review · Reviewer_qkCr · 2024-10-24

**Soundness:** 3
**Presentation:** 2
**Contribution:** 3
**Rating:** 6
**Confidence:** 4

**Summary:**

This paper proposes PolyNet, which aims to reinforce the exploration training and testing of Nueral CO methods. In particular, a representive NCO method POMO is used as the playground in this paper. The idea behind this paper is quite simple: adding a neural network module which contains MLP layers besides the  main structure of POMO, termed PolyNet layer. PolyNet layer  provide exploration ability by letting its input as the node embeddings and a binary context coding. Using different binary context codings, the output of PolyNet layer add different information to the output of POMO hidden layers. By training on moderate instances, PolyNet show state-of-the-art performance on several CO problems (#nodes < 300). Besides, the authors demonstrate the exploration ability of PolyNet during the training and testing is different from the common practice in existing works: forcing diverse first action selection.

**Strengths:**

The idea behind PolyNet is simple yet effective. The methodology part is easy-to-follow. The experimental results for performance comparison on different CO problems are solid and convincing. The ablation studies that explain the solution diversity in PolyNet are with good demonstration.

**Weaknesses:**

There are two major weaknesses:

a) Related works are not well-organized. As a important part to let readers aware the background, the motivation and the priority of this paper, Neural CO methods should be organized in a way that these points are clearly outlined.

b) Fow now, the scale of the CO problems PolyNet could address is no more than 300, which makes me curious about the performance of PolyNet on larger instances (#nodes > > 300). From my angle, the exploration ability is especially needed in large scale problems.

**Questions:**

See weaknesses.

---

> ### Author Response · Authors · 2024-11-15
>
> Thank you very much for reviewing our paper and providing helpful feedback! We are happy that you find the performance comparison on different CO problems solid and convincing.
>
> Please let us address your two remaining weaknesses.
>
> > a) Related works are not well-organized. As a important part to let readers aware the background, the motivation and the priority of this paper, Neural CO methods should be organized in a way that these points are clearly outlined.
>
> We agree that giving a good overview on the related work is of critical importance. As such we are very motivated to address this weakness and improve the literature review. Could you kindly provide some more details on how we can improve the related work section? Do you think that the related work section does not provide enough details? Are there works that we missed to include? Is the high-level structure of using 3 paragraphs (NCO, Diversity mechanisms in RL, Diversity mechanisms in neural CO) insufficient?
>
> We are confident that we will be able to improve the related work section during the rebuttal based on your feedback!
>
> > b) Fow now, the scale of the CO problems PolyNet could address is no more than 300, which makes me curious about the performance of PolyNet on larger instances (#nodes > > 300). From my angle, the exploration ability is especially needed in large scale problems.
>
> The version of PolyNet presented in this paper is not well suited to scale to very large instances without additional modifications, however our approach is nonetheless a significant contribution in its current form despite this limitation for two reasons: 1) A large number of real-world scenarios exit that require solving CVRP(TW) instances with fewer than 500 nodes. In fact, even more recent state-of-the-art, hand-crafted approaches from the operations research literature [1, 2] focus only on problems with 100 to 1000 nodes. On these medium-sized problems, our approach offers state-of-the-art performance and outperforms all other machine learning-based approaches. 2) Going forward, our method can be integrated into divide-and-conquer approaches that divide larger instances into smaller subproblems (see e.g., [3]). For these approaches, our method could function as a powerful subsolver, resulting in improved performance even on very large-scale instances.
>
> [1] Vidal, T. (2022). Hybrid genetic search for the CVRP: Open-source implementation and SWAP* neighborhood. Computers & Operations Research, 140, 105643. \
> [2] Christiaens, J., & Vanden Berghe, G. (2020). Slack induction by string removals for vehicle routing problems. Transportation Science, 54(2), 417-433. \
> [3] Ye, H., Wang, J., Liang, H., Cao, Z., Li, Y., & Li, F. (2024, March). Glop: Learning global partition and local construction for solving large-scale routing problems in real-time. In Proceedings of the AAAI Conference on Artificial Intelligence (Vol. 38, No. 18, pp. 20284-20292).

---

> > ### Comment · Reviewer_qkCr · 2024-11-19
> > **reply**
> >
> > a) For the related work part, the three paragraphs presented in the paper now just tell readers what works have been proposed in recent years. The limitation of them, as well as the motivation of this paper to address these limitations, is more important to help readers understand the significance of this paper.
> >
> > b) I expect an experiment integrating PolyNet into a divide-and-conquer framework for solving larger scale problems (Although I understand such development might be difficult and complex). Besides, given the rapid development of NCO, I am really confused about the significance of PolyNet, does it provide any insights for the further NCO research? If the large scale divide-and-conquer experiment could be conducted, some academic superiority might be revealed.

---

> > > ### Author Response · Authors · 2024-11-20
> > > **Comment 1/2**
> > >
> > > > For the related work part, the three paragraphs presented in the paper now just tell readers what works have been proposed in recent years. The limitation of them, as well as the motivation of this paper to address these limitations, is more important to help readers understand the significance of this paper.
> > >
> > > Thank you for providing additional clarification. We have improved the literature review based on your feedback putting a bigger emphasize on highlighting the limitations of earlier works and how PolyNet addresses these limitations. Please have a look at the updated version of our manuscript.
> > >
> > > > I expect an experiment integrating PolyNet into a divide-and-conquer framework for solving larger scale problems (Although I understand such development might be difficult and complex). Besides, given the rapid development of NCO, I am really confused about the significance of PolyNet, does it provide any insights for the further NCO research? If the large scale divide-and-conquer experiment could be conducted, some academic superiority might be revealed.
> > >
> > > Integrating PolyNet into a divide-and-conquer framework and conducting the required training of new models is unfortunately not feasible in the remaining rebuttal phase. However, we want to emphasize the significance of PolyNet on small and medium-scale problems and the relevancy of these problems in general.
> > >
> > > **Significance** PolyNet is significant because it outperforms all machine learning based approaches on problems with <= 300 nodes. This performance improvement is achieved by learning diverse strategies with a single model. The insights that we provide to the NCO community in this context are: 1) How diverse solution generation with a single decoder can be achieved. 2) The observation that diverse solution generation significantly improves performance. 3) That our diversity mechanism supersede the diversity mechanism introduced in the POMO method (where diverse first moves are forced) and can be used for more complex problems like the CVRPTW where the POMO diversity mechanism does not seamlessly work. Future works in in NCO field can use the presented concepts to enable better performance on more complex routing problems. As such, PolyNet is a significant advancement of the state-of-the-art in NCO.

---

> > > > ### Author Response · Authors · 2024-11-20
> > > > **Comment 2/2**
> > > >
> > > > **Small/medium-scale problems** The performance improvements of PolyNet on instances with less than 300 nodes are significant because these instances do frequently occur in practical applications and are still often considered in the literature. With PolyNet, we aim to make a step towards the practical application of NCO methods. We do this by focusing on more complex problems like the CVRPTW which are difficult to solve even with <300 customers.
> > > > To underscore that problems with less than <300 customers are still relevant today, we provide some very recent publications on VRP variants in the following together with the considered instance sizes:
> > > >
> > > > - Zhang, Jie, Yanfeng Li, and Zhaoyang Lu. "Multi-period vehicle routing problem with time windows for drug distribution in the epidemic situation." Transportation Research Part C: Emerging Technologies 160 (2024): 104484.
> > > > Considered sizes: 30-50 nodes
> > > > - Cataldo-Díaz, Cristian, Rodrigo Linfati, and John Willmer Escobar. "Mathematical models for the electric vehicle routing problem with time windows considering different aspects of the charging process." Operational Research 24.1 (2024): 1.
> > > > Considered sizes: 5-15 customers (plus nodes representing recharging stations)
> > > > - Lera-Romero, Gonzalo, Juan José Miranda Bront, and Francisco J. Soulignac. "A branch-cut-and-price algorithm for the time-dependent electric vehicle routing problem with time windows." European Journal of Operational Research 312.3 (2024): 978-995.
> > > > Considered sizes: 25-100 customers
> > > > - Huang, Nan, et al. "An exact algorithm for the multi-trip vehicle routing problem with time windows and multi-skilled manpower." European Journal of Operational Research (2024).
> > > > Considered sizes: The "large-sized instances" used in this paper comprise 25, 40, 50 customers (see beginning of Section 6.2.2)
> > > > - Jolfaei, Ali Aghadavoudi, and Mahdi Alinaghian. "Multi-depot vehicle routing problem with roaming delivery locations considering hard time windows: Solved by a hybrid ELS-LNS algorithm." Expert Systems with Applications 255 (2024): 124608.
> > > > Considered sizes: The "large instances" used in Section 5.4 of the paper comprise 48 to 288 customers.
> > > >
> > > >
> > > > All papers above have been published this year in highly-ranked journals and consider small instances. The papers have not been cherry-picked. We searched for "vehicle routing problem with time windows" limiting the search to only recent papers and focused on papers in highly-ranked venues. ALL publications that we looked at did not consider problems with more than 300 customers.
> > > >
> > > > In addition to the OR-focused papers above, there are countless ML-based methods that focus on problems with less than 300 nodes, e.g.,
> > > >
> > > > - Kim, Hyeonah, et al. "Symmetric Replay Training: Enhancing Sample Efficiency in Deep Reinforcement Learning for Combinatorial Optimization." arXiv preprint arXiv:2306.01276 (2023).
> > > > - Ma, Yining, Zhiguang Cao, and Yeow Meng Chee. "Learning to search feasible and infeasible regions of routing problems with flexible neural k-opt." Advances in Neural Information Processing Systems 36 (2024).
> > > > - Chalumeau, Felix, et al. "Combinatorial optimization with policy adaptation using latent space search." Advances in Neural Information Processing Systems 36 (2023): 7947-7959.
> > > >
> > > > The problems considered in OR journals, are more complex than the standard CVRP considered in most ML papers. PolyNet is relevant because it makes an important step towards solving these more complex problems by introducing a diversity mechanism that does not require certain symmetries in the solution space and by using an autoregressive construction process that does allow modeling much more complex problems.
> > > >
> > > > We agree with the reviewer, though, that large scale problems do need exploration and we hope to investigate them in future work.

---

> > > > > ### Comment · Reviewer_qkCr · 2024-11-23
> > > > > **Reply**
> > > > >
> > > > > The related works part looks better (yet not ideal) now, thanks for the authors' efforts. Besides, I acknowledge the significance of PolyNet in small instances and hope the authors could update the discussion into their paper. For now, however, I lean to maintain my score and wait for other reviewers' opinions.

---

> > > > > > ### Author Response · Authors · 2024-12-02
> > > > > >
> > > > > > Thank you again for your response. We have uploaded a new version of the paper that includes a new paragraph in the introduction discussing the significance of small instances: \
> > > > > > "PolyNet is an autoregressive construction method, which makes a direct application to large-scale routing problems computationally impractical. However, small- and medium-scale routing problems, which often involve numerous constraints (e.g., customer time windows), are frequently encountered in practice. This is reflected in the current focus of the operations research community, which prioritizes smaller problems with numerous complex constraints (e.g., Zhang et al. (2024); CataldoDíaz et al. (2024); Lera-Romero et al. (2024); Jolfaei & Alinaghian (2024)). With PolyNet, we aim to take a step toward addressing these more complex problems by enabling diverse solution generation without relying on the symmetries characteristic of simpler routing problems."
> > > > > >
> > > > > > >  For now, however, I lean to maintain my score and wait for other reviewers' opinions.
> > > > > >
> > > > > > We understand that you wanted to wait the responses of the other reviewers. However, given that the discussion phase is very shortly coming to an end, we hope that you reconsider your score based on the current information available and the changes that we made to the paper. Thank you once again for providing helpful feedback and engaging in a discussion.

---

> > > > > > > ### Comment · Reviewer_qkCr · 2024-12-02
> > > > > > >
> > > > > > > Dear authors of PolyNet, I change my score from boardline reject to boardline accept. The discussion we have made, should be included into this paper. Good Luck.

---

> > > > > > > > ### Author Response · Authors · 2024-12-02
> > > > > > > >
> > > > > > > > Thank you! We will include the discussion in the final version of the paper.

---

### Official Review · Reviewer_BPFA · 2024-10-31

**Soundness:** 2
**Presentation:** 3
**Contribution:** 3
**Rating:** 6
**Confidence:** 4

**Summary:**

This study, grounded in the POMO framework, introduces a straightforward yet impactful component: PolyNet. The component takes as input the concatenation of a set of binary vectors and the output from the decoder's multi-head attention layers. After processing through a series of linear layers and activation functions, the output is directly added to the output of the decoder's multi-head attention layers, thereby directly influencing the probability distribution of node selection. Analysis from the paper indicates that the newly added module helps to enhance solution diversity without the need for mandatory starting point selection as in POMO. Additionally, the authors assert that PolyNet can be integrated with EAS, and significant performance improvements can be achieved by updating only the newly added modules.

**Strengths:**

1. The authors have clearly articulated the framework and implementation of the method, with smooth writing.
2. The tables and figures are presented clearly, and the experiments are comprehensive.
3. Testing PolyNet on various combinatorial optimization problems (TSP, CVRP, CVRPTW, FFSP) demonstrates the good generality of the method.
4. (As claimed by the authors) their method performs exceptionally well on instances of scale 100, 200, and 300 (Tables 1-3).

**Weaknesses:**

PolyNet can enhance the diversity and optimality of solution sets, but the authors seem to lack an in-depth discussion on the principles behind the additional layers of PolyNet. This leaves me somewhat puzzled. The training process described in the paper adopts a method similar to Poppy, and the added linear layers, activation functions, and residual connections are common components in the Transformer architecture. The motivation for significant improvements by merely concatenating an additional binary array is not very clear.

**Questions:**

1. What is the design motivation behind the additional concatenated binary array $v$? Why can it significantly enhance diversity and optimality? If it were replaced with a column of random numbers, or if random perturbations were directly added to the input of the "added layer," would the same effect be achieved?

2. In Appendix Figure 6, a comparison is made between "PolyNet w/o Added Layers" and "POMO with Added Layers." It would be better if a "POMO" without additional additions could be included for comparison. My question is, why does "PolyNet w/o Added Layers" perform significantly better than "POMO with Added Layers"? Is it due to differences in training methods? If so, it would be best to compare POMO with the new training method.

3. Which contributes more significantly, the training method or the new structure?

4. In combinatorial optimization, there is a class of problems specifically focused on diversity optimization, where multiple solutions need to be found simultaneously. Relevant papers include “Computing Diverse Shortest Paths Efficiently: A Theoretical and Experimental Study” and “A Niching Memetic Algorithm for Multi-Solution Traveling Salesman Problem.” The work presented in this paper seems particularly suited for such scenarios, and I hope to see more discussion on this topic.

---

> ### Author Response · Authors · 2024-11-20
>
> Thank you for reviewing our paper and providing helpful feedback! We are happy that you like the writing and presentation of the paper and the performance of our proposed method.
>
> Please let us address your remaining questions.
>
>
> > What is the design motivation behind the additional concatenated binary array? Why can it significantly enhance diversity and optimality?
>
> Our key idea is to learn multiple, different construction strategies with a single neural network. To achieve this we condition the output of the network on the additional binary vector and train it so that different binary vectors result in the generation of different solutions. The results shown in Fig. 3 demonstrate that this indeed works. When sampling 100 solutions to a single instance with PolyNet the generated solutions are more diverse than those sampled via POMO, and provide better performance ("useful diversity").
>
>
> > If it were replaced with a column of random numbers, or if random perturbations were directly added to the input of the "added layer," would the same effect be achieved?
>
> This would achieve a similar effect. It is important that the model is given some input additional to the instance representation that it can condition its output on. The form of this additional input and how it is inserted does not seem that important. However, we observe experimentally that the additional layers that process the additional input do lead to significant performance improvements in Figure 6.
>
> > In Appendix Figure 6, a comparison is made between "PolyNet w/o Added Layers" and "POMO with Added Layers." It would be better if a "POMO" without additional additions could be included for comparison.
>
> Thank you for the suggestion. We have uploaded a new version of the paper that includes a comparison to POMO without additional layers. POMO without the additional layers performs slightly worse than the modified version with an additional layer.
>
> > My question is, why does "PolyNet w/o Added Layers" perform significantly better than "POMO with Added Layers"? Is it due to differences in training methods? If so, it would be best to compare POMO with the new training method.
>
> The added layer only has a small impact on performance. The main performance difference originates from the training method (including the additional input vector) and free first move selection. Note that the input vector is required for the training method to work, so that diverse strategies can be learned. Figure 6 now includes a comparison to the original POMO.
>
> > Which contributes more significantly, the training method or the new structure?
>
> As discussed above, the training method (including the additional vector input) is significantly more important. As shown in Figure 6, PolyNet without additional layers only works slightly worse than the PolyNet version with additional layers. Despite their small impact, we use the additional layers because they have an insignificant impact on runtime and are very easy to implement.
>
> > In combinatorial optimization, there is a class of problems specifically focused on diversity optimization, where multiple solutions need to be found simultaneously. Relevant papers include “Computing Diverse Shortest Paths Efficiently: A Theoretical and Experimental Study” and “A Niching Memetic Algorithm for Multi-Solution Traveling Salesman Problem.” The work presented in this paper seems particularly suited for such scenarios, and I hope to see more discussion on this topic.
>
> Thank you for bringing this area of research to our attention. We have updated the related work section of our paper to include a discussion on these works. It further confirms the importance of useful diversity for effective search strategies. Please see that updated version of our manuscript.

---

> > ### Author Response · Authors · 2024-11-26
> >
> > Dear Reviewer BPFA,
> >
> > Thank you again for your valuable feedback on our paper!
> >
> > As the deadline for PDF modifications is tomorrow, we wanted to check if we’ve fully addressed all your concerns or if there are any remaining issues. We are keen to improve our paper further, and we truly appreciate your insights in helping us achieve this.
> >
> > We hope that our clarifications and updates warrant a reassessment of your current score, and we are grateful for your consideration.
> >
> > Best regards,\
> > The PolyNet Authors

---

> > > ### Author Response · Authors · 2024-12-02
> > >
> > > Dear Reviewer BPFA,
> > >
> > > Thank you again for reviewing our paper!
> > >
> > > As today is the last day that reviewers can post comments, we wanted to ask if you have any remaining questions. We hope that we were able to address most of your concern in our discussion. If this is not the case, please let us know.
> > >
> > > We also hope that you reconsider you current score based on our discussion and the changes made to the paper. Thank you once again for providing helpful feedback.
> > >
> > > Best regards, \
> > > The PolyNet Authors

---

> > > > ### Comment · Reviewer_BPFA · 2024-12-03
> > > > **keep my score**
> > > >
> > > > Thank you for the rebuttal and revisions. I appreciate the clarification on PolyNet's approach and the experimental results. I have no further comments. Based on my evaluation, while the proposed method shows promising results, I feel the idea of improving exploration does not offer a particularly groundbreaking contribution. As a result, I would like to maintain my original score of 6.

---

### Official Review · Reviewer_rgaA · 2024-10-31

**Soundness:** 2
**Presentation:** 3
**Contribution:** 1
**Rating:** 3
**Confidence:** 5

**Summary:**

To facilitate learning complementary solution strategies, this paper avoids forcing the initial action step. Instead, it enhances exploration by embedding it into the decoder as a bit vector. While experiments demonstrate performance improvements, the approach lacks novelty.

**Strengths:**

1. The paper is well-structured, presenting its content in a clear manner.
2. A wide range of experiments are conducted to assess solution diversity.

**Weaknesses:**

1. The method of inserting additional vectors in the decoder is very similar to the idea of COMPASS [1], lacking sufficient innovation to significantly advance the field.
2. The description of bit vectors is minimal, and there is no detailed analysis of how varying vector representations might impact performance.
3. The fairness of comparing COMPASS with PolyNet+EAS in an experimental context is questionable.
4. In [1], there is a big difference between COMPASS and EAS runtimes, but in this paper, PolyNet+EAS time is similar to COMPASS, please specify the reason.

[1] Combinatorial optimization with policy adaptation using latent space search. NeurIPS 2023.

**Questions:**

1. The advantages of using bit vectors as supplementary vectors for insertion should be thoroughly analyzed, particularly in comparison to previous continuous potential vectors.
2. In the experimental section, a more comprehensive explanation and analysis of performance concerning runtime are necessary.

---

> ### Author Response · Authors · 2024-11-20
>
> Thank you for reviewing our paper and providing helpful feedback! We are happy that you like the presentation and the experimental evaluation of our paper.
>
> Please let us address your remaining concerns.
>
> > 1. The method of inserting additional vectors in the decoder is very similar to the idea of COMPASS [1], lacking sufficient innovation to significantly advance the field.
>
> The COMPASS method is a high-quality contribution to the field, but we emphasize that PolyNet is quite different, even though they both insert a vector into the decoder. PolyNet aims to learn discrete strategies, while COMPASS learns a continuous latent spaces that can be searched by a continuous search method like CMA-ES. Most importantly, PolyNet can generate solutions quickly (e.g., 4 minutes for 10,000 TSP instances) while COMPASS is designed for longer runtimes (e.g., 2 hours for 10,000 instances). While PolyNet can also be combined with EAS for longer search runs, our main focus is on generating solutions quickly.
>
> > 2. The description of bit vectors is minimal, and there is no detailed analysis of how varying vector representations might impact performance.
>
> We believe that the bit vectors can be replaced with other discrete tokens without a major impact on performance. What matters is that the network has some kind of input that it can condition its output on. We have chosen bit vectors as they are simple to implement and require no extra normalization. In preliminary experiments, we also evaluated one-hot encoding instead of binary encoding for the bit vector (i.e., learning 128 strategies would then require a bit vector of size 128 in contrast to a bit vector of size 7), but we found no major differences. Are there any specific representations that you have in mind? Overall, we are confident that the exact form of the “discrete input token” is not very relevant, simply because the network can learn to transform the bit vector representation into a more beneficial representation on its own.
>
> > 3. The fairness of comparing COMPASS with PolyNet+EAS in an experimental context is questionable.
>
> Why do you consider this an unfair comparison? COMPASS has been designed with longer runtimes in mind and uses CMA-ES to provide high-level search guidance. PolyNet without EAS lacks such a search component. We could emphasize more strongly in the paper, that EAS increases the runtime due to the needed parameter updates, if this is your main concern.
>
> > 4. In [1], there is a big difference between COMPASS and EAS runtimes, but in this paper, PolyNet+EAS time is similar to COMPASS, please specify the reason.
>
> Note that the OpenReview version of the COMPASS method contains incorrect runtimes for EAS as described on the Github page of COMPASS. The recent arxiv version contains the corrected values.
>
> There are still some differences to the runtimes reported in our work. We can not provide a definite explanation because we do not have access to all experiment protocols of the COMPASS paper or the same hardware. We can only speculate that the difference originate from differences in the hardware used (we evaluate EAS on an A100 while the COMPASS authors use a TPU), differences in the implementation (we use PyTorch with FlashAttention while the COMPASS authors use JAX). Due to these differences in the experimental settings the runtimes can not easily compared. To ensure a fair comparison we follow earlier work (including the COMPASS paper) and limit the search by the number of generated solution.
>
>
> > The advantages of using bit vectors as supplementary vectors for insertion should be thoroughly analyzed, particularly in comparison to previous continuous potential vectors.
>
> As outlined above, we did not found significant performance differences between different vector representations in preliminary experiments. It should also be possible to use continuous vectors instead of bit vectors without an impact in performance.
>
> > In the experimental section, a more comprehensive explanation and analysis of performance concerning runtime are necessary.
>
> The runtime of the methods is highly influenced by the quality of their implementation, the chosen framework, and the hardware used. Therefore, small differences in runtime should not be overinterpreted. We can highlight this more explicitly in the paper and underscore that EAS incurs additional runtime overhead due to the required parameter updates.

---

> ### Comment · Reviewer_rgaA · 2024-11-24
>
> Thanks to the authors' efforts in the rebuttal. I am inclined to keep my score, since the methodological contribution of this paper appears to be quite limited, given the following reasons:
> 1. While the use of discrete vectors is emphasized in this work, the COMPASS paper has already discussed discrete vectors, as noted, "There are several ways to define the latent space, and the simplest approach is to use a set of N one-hot encoded vectors."
> 2. As stated by the authors in their response, "PolyNet aims to learn discrete strategies, while COMPASS learns a continuous latent spaces that can be searched by a continuous search method like CMA-ES", alongside with the claim that "It should also be possible to use continuous vectors instead of bit vectors without an impact in performance". These two points, when considered together, suggest that the choice of discrete vectors is not particularly significant.
> 3. The authors emphasize that "our main focus is on generating solutions quickly", which is a limited contribution in my opinion.

---

> > ### Author Response · Authors · 2024-11-25
> >
> > Thank you for your response, we appreciate your engagement with our work.
> >
> > You are right that COMPASS and PolyNet condition solution generation on an additional vector input. Whether these are bit vectors or continuous vectors should not have a big impact on performance for PolyNet, although a discrete space vector might be hard to explore in COMPASS.
> >
> > However, there is one very important difference between our work an COMPASS that we'd like to point out. COMPASS aims to learn a latent space that contains a high number of different solutions (e.g., millions of different solutions). COMPASS hence uses a search method that navigates through this latent space. This search method is necessary because the latent space maps to so many solutions that not all solutions can be explored. In contrast, PolyNet does not aim to create a latent space. PolyNet only considers a fixed amount (128 in most experiments) of discrete vectors during solution generation. Since there are only 128 different vector that need to be explored, we do not need a search method. Instead, the model can rely on the fact that during training and testing all 128 vector (i.e, strategies) can take a shot at generating the best solutions.
> >
> > **In summary, the dichotomy between COMPASS and Polynet can be thought of as follows: COMPASS creates a searchable fitness landscape, PolyNet creates a finite set of discrete strategies.**
> >
> > This difference is especially noticeable for fast solution generation. PolyNet can be used to quickly (in the fraction of a second) generate high quality solutions to an instance (by sampling one solution per learned strategy), whereas COMPASS requires searching through a high-dimensional latent space with high-level search method which results in much longer runtimes.
> >
> > > The authors emphasize that "our main focus is on generating solutions quickly", which is a limited contribution in my opinion.
> >
> > We appreciate the reviewer’s perspective and agree that there are use cases where decision-makers can afford longer runtimes. However, there are also many practical scenarios where fast solution generation is important, such as real-time decision-making and interactive, human-in-the-loop decision support. Importantly, the focus on fast solution generation aligns with the prevailing trend in the machine learning community, where learning-based methods are designed to solve tens of thousands of instances within minutes. We respectfully suggest that emphasizing fast solution generation is not a limitation of our contribution but rather reflects an important and widely recognized priority in the machine learning community.
> >
> > We hope this perspective helps highlight the contribution and relevance of our work. While PolyNet and COMPASS share some similarities, the key idea in our approach is fundamentally different, as we focus on learning a predefined set of strategies rather than latent space. Additionally, our work incorporates smaller yet meaningful innovations, such as deliberately not using the long-standing "forced first move" technique introduced in the POMO paper, which allows for a more generalized and flexible solution approach.
> >
> > Given these contributions, we kindly encourage the reviewer to reassess their evaluation. We believe the novelty and practical value of our approach extend beyond what is reflected in a score of 3.

---

> > > ### Author Response · Authors · 2024-12-02
> > >
> > > Dear Reviewer rRFp,
> > >
> > > Thank you again for reviewing our paper and providing helpful feedback!
> > >
> > > As today is the last day that reviewers can post comments, we wanted to ask if you have any remaining questions. We hope that we were able to address most of your concern in our discussion. If this is not the case, please let us know.
> > >
> > > There are certain similarities between COMPASS and PolyNet and we understand why you initially came to the conclusion that our paper has a limited contribution. We hope that we were able to convince you that there are certain critical differences between both methods. For example, PolyNet is able to generate solutions in the fraction of a second, which is the standard evaluation scenario considered in the ML literature. The exceptional performance of PolyNet in such settings makes it highly relevant to the NCO community.
> > >
> > > We hope that you reconsider you current score based on our discussion and the changes made to the paper.
> > >
> > > Best regards, \
> > > The PolyNet Authors

---

> > > > ### Comment · Reviewer_rgaA · 2024-12-03
> > > >
> > > > Thank you for your thoughtful response. I acknowledge that this paper does make a certain contribution. However, given the similarities between PolyNet and COMPASS, I am concerned that its level of innovation may not meet the threshold of ICLR. Thus, I maintain my rating.

---

### Official Review · Reviewer_rRFp · 2024-11-04

**Soundness:** 3
**Presentation:** 3
**Contribution:** 3
**Rating:** 6
**Confidence:** 4

**Summary:**

PolyNet introduces a novel approach to enhancing exploration in neural combinatorial optimization (CO). Unlike traditional methods that enforce diversity via handcrafted rules, PolyNet employs a single-decoder model that inherently learns multiple strategies, facilitating more effective search and solution quality. The paper demonstrates PolyNet's superior performance across several benchmark CO problems, such as TSP, CVRP, and CVRPTW, outperforming state-of-the-art methods both in speed and solution quality.

**Strengths:**

- The use of a single-decoder model that can learn multiple strategies simplifies the training pipeline and reduces computational overhead compared to multi-decoder systems.

- The model's training schema, which emphasizes inherent diversity, makes it adaptable to various CO problems beyond simple routing tasks.

- The authors provide useful insights into the impact of their design choices, such as the effectiveness of not forcing diverse first moves.

**Weaknesses:**

- While starting from pre-trained models boosts training efficiency, this reliance could limit applicability in cases where such pre-training is not feasible or available.

- While PolyNet demonstrates strong results in routing tasks, its effectiveness in other CO problem domains (e.g., scheduling or knapsack) has not been deeply explored.

- The paper does not discuss how PolyNet handles instances where input data is noisy or incomplete, a common occurrence in practical applications.

- The impact of different hyperparameter settings on model performance is not fully analyzed, which could be critical for real-world applications requiring fine-tuning.

**Questions:**

1. How would PolyNet perform when extended to non-routing CO problems, such as job scheduling or knapsack problems?

2. Can you elaborate on the potential methods for improving PolyNet’s scalability to handle larger problem instances?

3. What are the specific limitations of PolyNet’s inherent diversity mechanism when applied to problem instances with different statistical properties?

---

> ### Author Response · Authors · 2024-11-20
> **Comment 1/2**
>
> Thank you very much for reviewing our paper and providing helpful feedback! We are happy that you point us many of the core components of our work as strengths and find that we provide useful insights into the impact of our design choices.
>
> Please let us address your remaining concerns and your questions.
>
> > While starting from pre-trained models boosts training efficiency, this reliance could limit applicability in cases where such pre-training is not feasible or available.
>
> PolyNet can also be trained from scratch, but starting from a pre-trained model indeed boosts training efficiency. The same is true for virtually all deep learning models. We believe that pre-trained models will often already be available in real-world scenarios. If no pre-trained models are available, users can start training from scratch or try self-supervised pre-training methods.
>
> > While PolyNet demonstrates strong results in routing tasks, its effectiveness in other CO problem domains (e.g., scheduling or knapsack) has not been deeply explored.
>
> We evaluate PolyNet on a scheduling problem (the flexible flow shop problem (FFSP)) in Appendix A. On this problem, PolyNet shows improvement in line with the improvements observed on the routing problems, demonstrating that it is applicable to a wider range of problems.
>
> > The paper does not discuss how PolyNet handles instances where input data is noisy or incomplete, a common occurrence in practical applications.
>
> We agree that noisy input data is a common occurrence in practical applications and that methods that address this (e.g., stochastic optimization methods) deserve more attention. However, almost all NCO paper currently focus on deterministic optimization problems and assume complete information. We intend to evaluate PolyNet on stochastic problems in future work. The evaluation on stochastic problems requires a different experimental setup and a more in-depth discussion that would be beyond the scope of this paper.
>
> > The impact of different hyperparameter settings on model performance is not fully analyzed, which could be critical for real-world applications requiring fine-tuning.
>
> One of the benefits of PolyNet is that it does not require hyperparameter tuning for good performance and we also did not conduct any hyperparameter tuning. The impact of the main hyperparameter $K$ (i.e., the number of learned strategies) is shown in Figure 3, with higher-values resulting in better performance. However, the selection of $K$ is effectively limited by the available GPU memory. All other parameters like the batch size and learning rate are selected based on experience with the POMO model to enable stable training. The performance improvements that can be improved by tuning these values are not worth the additional computation costs associated with tuning these values.
>
> If there is any hyperparamter that you are particularly interested in, please let us know!

---

> > ### Author Response · Authors · 2024-11-20
> > **Comment 2/2**
> >
> > > How would PolyNet perform when extended to non-routing CO problems, such as job scheduling or knapsack problems?
> >
> > See Appendix A and our response above!
> >
> > > Can you elaborate on the potential methods for improving PolyNet’s scalability to handle larger problem instances?
> >
> > The version of PolyNet presented in this paper is not well suited to scale to very large instances without additional modifications. There are two reasons why we believe that our approach in its current form is a significant contribution despite this limitation: 1) A large number of real-world scenarios exist that require solving CVRP(TW) instances with fewer than 500 nodes. In fact, even more recent state-of-the-art, hand-crafted approaches from the operations research literature [1, 2] focus only on problems with 100 to 1000 nodes, many of which contain 300 or less nodes. On these medium-sized problems, our approach offers state-of-the-art performance and outperforms all other machine learning-based approaches. 2) Going forward, our method can be integrated into divide-and-conquer approaches that divide larger instances into smaller subproblems (see e.g., [3]). For these approaches, our method could function as a powerful subsolver, resulting in improved performance even on very large-scale instances.
> >
> > Please also see our response to reviewer qkCr on this matter.
> >
> > [1] Vidal, T. (2022). Hybrid genetic search for the CVRP: Open-source implementation and SWAP* neighborhood. Computers & Operations Research, 140, 105643.
> > [2] Christiaens, J., & Vanden Berghe, G. (2020). Slack induction by string removals for vehicle routing problems. Transportation Science, 54(2), 417-433.
> > [3] Ye, H., Wang, J., Liang, H., Cao, Z., Li, Y., & Li, F. (2024, March). Glop: Learning global partition and local construction for solving large-scale routing problems in real-time. In Proceedings of the AAAI Conference on Artificial Intelligence (Vol. 38, No. 18, pp. 20284-20292).
> >
> > > What are the specific limitations of PolyNet’s inherent diversity mechanism when applied to problem instances with different statistical properties?
> >
> > We evaluate PolyNet's ability to generalize to instances with different properties by using models trained on instances with 100 and 200 customers to solve instances with 150 and 300 customers, respectively. PolyNet shows excellent generalization capabilities in this setting, beating all other methods in this experiment. PolyNet's ability to learn diverse strategies actually improves its generalization capability over other methods.
> >
> > We hope that this answers your question. If we have misinterpreted it, please let us know.

---

> > > ### Author Response · Authors · 2024-11-26
> > >
> > > Dear Reviewer rRFp,
> > >
> > > Thank you again for your valuable feedback on our paper!
> > >
> > > As the deadline for PDF modifications is tomorrow, we wanted to check if we’ve fully addressed all your concerns or if there are any remaining issues. We are keen to improve our paper further, and we truly appreciate your insights in helping us achieve this.
> > >
> > > We hope that our clarifications and updates warrant a reassessment of your current score, and we are grateful for your consideration.
> > >
> > > Best regards,\
> > > The PolyNet Authors

---

> ### Comment · Reviewer_rRFp · 2024-11-28
>
> Thanks for your response. You have addressed my concerns, and I'd like to keep my score.

---

> > ### Author Response · Authors · 2024-11-28
> >
> > Thank you for your response and for acknowledging that we have addressed all your concerns. Given that there are no remaining issues and you have highlighted several strengths of our paper, we feel that a score of 6 does not fairly reflect the provided feedback. In the interest of fairness, we kindly ask for further clarification or reconsideration of the score. We hope you can understand our position, as it is difficult to see why the paper is not considered 'good' (score 8) when there are several strengths and no outstanding concerns.

---

### Author Response · Authors · 2024-11-20
**Response to all reviewers**

Thank you once again for your constructive feedback!

We have uploaded a revised version of the paper that incorporates your suggestions. Specifically, we have restructured the literature review to better emphasize the limitations of prior works and clearly outline the distinctions of our approach. Additionally, we have extended Figure 6 to include results for the original POMO method without additional layers, demonstrating that PolyNet also outperforms the unmodified version of POMO.

If you have any further questions or concerns, we would be happy to address them.

---

### Meta-Review · Area_Chair_tADh · 2024-12-21

**Metareview:**

The paper proposes a new policy-based method for combinatorial optimization. The core idea is to form a $\pi(a | s, v)$ where:
  * $a$ is a possible problem solution (parameterized via the use of standard Pointer Network / Attention mechanisms)
  * $s$ is a representation of the initial problem conditions (e.g. city locations in Travelling Salesman)
  * $v$ is a diversity prompt, meant to ensure that the corresponding $a$ should be diverse and far away from other $a'$ from other $v'$.

Training is done via the Poppy (Grinsztajn et al) method, in which policy gradients are only backpropagated for the best solutions $(a, s, v)$ when varying $v$. Experiments are conducted over standard combinatorial optimization problems e.g. (TSP, CVRP, CVRPTW).

Overall, this paper builds on top of the Poppy regime, with the improvement being that there is now only a single decoder needed, with diversity based on "diversity prompts" $v$ rather than needing multiple separate decoder heads, making it more efficient in practice. While the results and execution are solid, one could argue that this is still fairly incremental work.

Given that most of the reviewers have voted to accept (see also my analysis of the issues raised by the reviewer who voted to reject), the decision is a poster accept.

**Additional Comments On Reviewer Discussion:**

The post-rebuttal reviewer scores are (3,6,6,6), making this paper borderline. Most of the reviewers agree that this paper is solid, although borderline.

Reviewer rgaA who gave the 3 makes the claim that this work is too similar COMPASS (Chalumeau et al.) are too similar. I read the COMPASS paper carefully, and the differences are subtle but nontrivial:
  * COMPASS is a policy $p(a | s, v)$ where $s$ is the initial problem description and $v$ is a latent vector used to prompt out a proposal $a$. This effectively leads to a mapping from a continuous space to the discrete space, and thus at inference time, one perform explicit multi-turn blackbox optimization over the new continuous space of $v$ (e.g. use CMA-ES).
  * PolyNet is a policy $p(a | s, v)$ similar to before, but the use-case is meant to be _zero shot_ (i.e. we expect to obtain a good solution $a$ immediately from sampling). Furthermore, the intention of $v$ is a "diversity prompt" rather than a bijective mapping between continuous and discrete space.

Basically, the differences are whether to use $\pi$ as a mapping between continuous and discrete spaces vs. using $\pi$ as a zero shot proposal with diversity prompts. These mechanics are indeed fundamentally different, and thus this would not be a valid point for rejection.

---

### Decision · Program_Chairs · 2025-01-22

Accept (Poster)